# A Design Approach for Simultaneous Cooperative Interception Based on Area Coverage Optimization

**Long Wang** [1,2], **Kai Liu** [3,*], **Yu Yao** [4] and **Fenghua He** [5]

1    China Airborne Missile Academy, Luoyang 471000, China; hitwanglong@163.com
2    Aviation Key Laboratory of Science and Technology on Airborne Guided Weapons, Luoyang 471000, China
3    School of Aeronautics and Astronautics, Dalian University of Technology, Dalian 116024, China
4    Control and Simulation Center, Harbin Institute of Technology, Harbin 150080, China; yaoyu@hit.edu.cn
5    School of Astronautics, Harbin Institute of Technology, Harbin 150080, China; hefenghua@hit.edu.cn
*    Correspondence: carsonliu@dlut.edu.cn

**Abstract:** In this paper, a design approach for simultaneous cooperative interception is presented for a scenario where the successful handover cannot be guaranteed by a single interceptor due to the target maneuver and movement information errors at the handover moment. Firstly, the concepts of the reachable interception area and predicted interception area are introduced, a performance index function is constructed, and the probability of a successful handover is described by considering the coverage of the predicted interception area. Taking the probability of successful handover as a constraint, the simultaneous cooperative interception design problem is formulated based on area coverage. Then, an area coverage optimization algorithm is presented to design the spatial distributions of the interceptors. In order to enhance the handover probability, a simultaneous cooperative interception design approach is proposed to obtain the number of interceptors and the corresponding spatial distributions. Finally, simulation experiments are carried out to validate the effectiveness of the proposed approach.

**Keywords:** cooperative interception; area coverage optimization; multiple interceptors

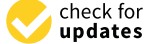



## 1. Introduction

For the interception of an invading high-speed target, the common engagement process can be divided into three phases, i.e., the boost phase, mid-course guidance phase, and terminal guidance phase. In the boost phase, the interceptor gains the velocity fast enough for the mid-course guidance phase. In the mid-course guidance phase, the interceptor flies to the predicted interception point, which is provided by the ground or space-based target tracking system. When the interceptor gets close enough to the target, the seeker system of the interceptor starts working and the terminal guidance begins, in which the interceptor flies to the target using proportional guidance law or augmented proportional guidance law [1]. A successful handover from mid-course guidance to terminal guidance is a necessary condition for the interceptor to hit the target, i.e., the miss distance caused by the target maneuver and movement information errors is less than the maximum distance that the interceptor can maneuver in the terminal guidance. However, in some interception scenarios, the target movement information cannot be obtained accurately, such as for a newly emerging hypersonic target in near space, so the resulted miss distance may be larger than the maximum maneuverable distance of an interceptor. In these circumstances, a successful handover cannot be guaranteed by a single interceptor; therefore, cooperative interception using multiple interceptors is gradually becoming a trend in the development of defense technologies.

For the state of the art of cooperative interception, existing research results mainly consist of task assignment and cooperative guidance law design. In [2], the interception of multiple high-speed targets was considered, and the kill probabilities of targets were taken

as the performance index function for the task assignment. In [3], a discrete particle swarm optimization algorithm was proposed to search for the optimal task assignment scheme. In [4], the task assignment was formulated based on game theory, and two negotiation mechanisms were proposed to search for the optimal assignment scheme. Considering the uncertainties of the target invading direction, a Salvo Enhanced No Escape Zone algorithm was proposed to allocate the task to multiple interceptors in [5], which increased the chances that the target was killed by at least one interceptor. In [6], the handover of two interceptors with simultaneous cooperative interception was considered and the locations of the interceptors were designed by maximizing the overall engagement envelopes, but the distributions of the target movement information errors were not taken into account. For the design of cooperative guidance law, the main purpose is to guarantee that all the interceptors encounter the target at the same time or at a certain angle with respect to each other [7–15], and the miss distance of each interceptor is zero with respect to the predicted interception point. However, when all the interceptors are aiming at the same predicted interception point at the handover moment, the maximum maneuverable distance may be less than all the miss distances of the interceptors with respect to the target, which means that all the interceptors are unable to hit the target in the terminal guidance. Thus, besides the coordination of the time and angle, it is necessary to optimize the spatial distribution of the predicted interception points of the interceptors so that there is at least one interceptor capable of hitting the target. According to the current research status of cooperative interception, most of the results are based on the condition that the target movement information is accurate, which is unrealistic in some engagement scenarios. When accurate information about the target is unavailable, only the possible positions of the target can be obtained. In order to ensure a successful interception, the number of interceptors should be sufficient and the area should be allocated to each interceptor so that the target area can be completely covered by the reachable areas of the interceptors. However, existing methods for the task assignment of many-on-many interception engagements cannot be used for the allocation of the target area. Therefore, it is necessary to study the cooperative interception design approach based on area coverage. Area coverage is an important research direction in the cooperative control of multiple agents, and many approaches have been developed for various kinds of coverage problems [16–21], such as environment monitoring, sweeping, search and rescue, and sensor node arrangements in wireless sensor networks, etc. However, a coverage approach is aimed at a clear application background and the interceptor is different from the agent, thus the existing coverage approaches cannot be directly applied to the cooperative interception design.

In this paper, considering the uncertainties of the target maneuver and movement information, the problem of cooperative interception is investigated. Taking both the miss distance and probability of successful handover into account, the spatial distributions and number of interceptors are designed for the handover of simultaneous cooperative interception. The remainder of this paper is organized as follows. In Section 2, the mathematical descriptions of the reachable area and predicted interception area are presented first, and considering the coverage of the predicted interception area, a performance index function for cooperative interception is constructed. Then, taking the probability of successful handover as a constraint, the cooperative interception problem is formulated based on area coverage. In Section 3, an area coverage optimization algorithm is proposed to optimize the spatial distribution of interceptors, then an approach for the cooperative interception problem is proposed based on the area coverage optimization algorithm. Simulations are carried out to illustrate the effectiveness of the proposed approach in Section 4. Conclusions and future work are presented in Section 5.

## 2. Problem Formulation

In this section, the descriptions of the reachable area of an interceptor and predicted interception area are given, then the cooperative interception design problem is formulated based on area coverage optimization.

### 2.1. Reachable Area of an Interceptor

Before the description of the reachable area, the movement model of an interceptor is presented, which is based on the following assumptions.

**Assumption 1** [22]: *The relative trajectories between an interceptor and the target can be linearized with respect to the initial line of sight, and the acceleration of an interceptor is perpendicular to the line of sight.*

**Assumption 2:** *The dynamics of the interceptor can be neglected.*

**Assumption 3:** *The control of an interceptor in the longitudinal plane and lateral plane can be decoupled from each other.*

The interception geometry between an interceptor and a target is shown in Figure 1, where $Oxyz$ is the initial line-of-sight frame. The origin $O$ is set to be the position of an interceptor at the initial moment, the $Ox$ axis is along the initial line of sight, the $Oy$ axis is perpendicular to *the Ox* axis and lies in the vertical plane containing the $Ox$ axis, and the $Oz$ axis is determined by the right-hand rule. The target and interceptor are denoted by T and I, respectively, and $[x_T, y_T, z_T]$ and $[x_I, y_I, z_I]$ are the positions of the target and interceptor. $V_T$ and $V_I$ denote the velocity of the target and interceptor, respectively.

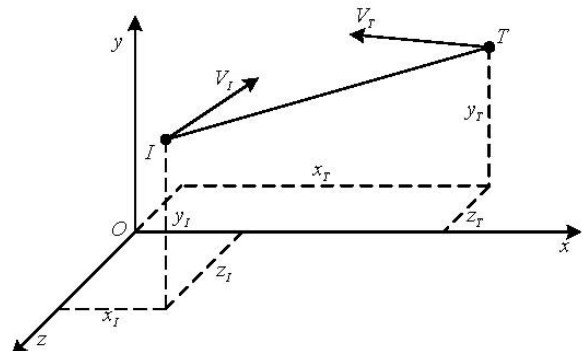

**Figure 1.** Engagement geometry between an interceptor and a target.

Based on Assumption 1, let $\boldsymbol{y}_I = \begin{bmatrix} y_I & z_I \end{bmatrix}^{\mathrm{T}}$ be the lateral movement states of an interceptor in the $Oxyz$ frame, then the movement model of an interceptor in the initial line-of-sight frame can be described as

$$\dot{\boldsymbol{y}}_I = \boldsymbol{A}_I \boldsymbol{y}_I + \boldsymbol{B}_I \boldsymbol{a}_I \tag{1}$$

where $a_I$ is the acceleration of an interceptor and

$$A_I = \begin{bmatrix} 0 & 0 & 1 & 0 \\ 0 & 0 & 0 & 1 \\ 0 & 0 & 0 & 0 \\ 0 & 0 & 0 & 0 \end{bmatrix}, \; B_I = \begin{bmatrix} 0 & 0 \\ 0 & 0 \\ 1 & 0 \\ 0 & 1 \end{bmatrix} \tag{2}$$

The acceleration of an interceptor is generally limited, so let $a_{\max}$ be the maximum value of the interceptor acceleration, and then the constraint set of $\boldsymbol{a}_I$ can be written as

$$U_I = \left\{ \boldsymbol{a}_I \middle| |a_{Iy}| \leq a_{\max}, \; |a_{Iz}| \leq a_{\max} \right\} \tag{3}$$

where $a_{Iy}$ and $a_{Iz}$ are the $x$ and $y$ components of $\boldsymbol{a}_I$, respectively.

Now, the description of the reachable area will be presented based on the dynamic model and input constraints. First, the following definitions are given.

**Definition 1:** *Predicted interception moment: under the condition that the control of an interceptor is zero, the moment when the distance between the interceptor and target reaches minimum is called the predicted interception moment, which is denoted by $t_e$. Assuming that the interceptor stays parallel approaching the target, then the predicted interception moment can be estimated as*

$$t_e = t + \frac{R_r(t)}{V_r(t)} \tag{4}$$

*where $R_r(t)$ and $V_r(t)$ are the distance and approaching speed between the interceptor and target, respectively.*

**Definition 2:** *Terminal point under zero control: when the control input of an interceptor is zero, i.e., $\boldsymbol{a}_T(t) = \boldsymbol{0}, t \in [t_0, t_e]$ , the point that the interceptor reaches at $t = t_e$ is called the terminal point under zero control, which is abbreviated to TPZC. Let $\boldsymbol{p}_I = [p_y, p_z]^{\mathrm{T}}$ be the coordinates of TPZC then, according to the movement model of the interceptor, the expression of $\boldsymbol{p}_I$ is*

$$\boldsymbol{p}_I = \boldsymbol{C}\boldsymbol{\Phi}_I(t_e, t_0)\boldsymbol{y}_I(t_0) \tag{5}$$

*where, $\boldsymbol{C} = \begin{bmatrix} \boldsymbol{I}_{2\times2} & \boldsymbol{0}_{2\times2} \end{bmatrix}$ and $\boldsymbol{\Phi}_I(t_e, t_0)$ is the state transition matrix of system (1), which is expressed as*

$$\boldsymbol{\Phi}_I = \begin{bmatrix} \boldsymbol{I}_{2\times2} & (t_e - t_0)\boldsymbol{I}_{2\times2} \\ \boldsymbol{0}_{2\times2} & \boldsymbol{I}_{2\times2} \end{bmatrix} \tag{6}$$

**Definition 3:** *Reachable area of an interceptor: under the control input $a_I(t)$ $\forall t \in [t_0, t_e]$, the set of points that the interceptor can reach at $t = t_e$ is called the reachable area of an interceptor, which is denoted by $\overline{\mathcal{M}}(t)$.*

Under the constraints of $\boldsymbol{a}_I$, the reachable area of an interceptor at $t = t_0$ can be expressed as

$$\overline{\mathcal{M}}(t_0) = \left\{ (y, z) \left| [y, z]^{\mathrm{T}} = \boldsymbol{C}\left( \boldsymbol{\Phi}_I(t_e, t_0)\boldsymbol{y}_I(0) + \int_{t=t_0}^{t_e} \boldsymbol{\Phi}_I(t_e, \tau)\boldsymbol{B}_I\boldsymbol{a}_I(\tau)d\tau \right), \boldsymbol{a}_I(\tau) \in U_I \right. \right\} \tag{7}$$

Let $d_0$ be the maximum distance that an interceptor can maneuver within the time interval $t \in [t_0, t_e]$, which can be described by

$$d_0 = \frac{1}{2}a_{\max}(t_e - t_0)^2 \tag{8}$$

Then according to Equations (3) and (7), the reachable area of an interceptor at $t = t_0$ can be rewritten as

$$\overline{\mathcal{M}}(t_0) = \left\{ (y, z) \left| |y - p_y| \le d_0, \ |z - p_z| \le d_0 \right. \right\} \tag{9}$$

*2.2. Predicted Interception Area*

Now, the definition of the predicted interception area is given, followed by its mathematical description.

**Definition 4:** *Predicted interception area: under the condition of target maneuver and movement information errors, the set of all possible locations that the target may appear at $t = t_e$ is called the prediction interception area, which is denoted by $\mathcal{R}$.*

Consider the following target movement model:

$$\dot{\boldsymbol{X}}_T(t) = \boldsymbol{f}_T(\boldsymbol{X}_T(t), u_T(t), t) \tag{10}$$

where $X_T(t)$ and $u_T(t)$ are the movement states and control input of the target, respectively.

If the target movement model is nonlinear, the analytical solutions of Equation (10) cannot be obtained directly and the location of the target at $t = t_e$ can be obtained by numerical calculation. Discretize the target movement model by time interval $\Delta T$, and the discrete movement model can be expressed as

$$X_T(k+1) = X_T(k) + \Delta T f_T(X_T(k), u_T(k), k) \tag{11}$$

It is assumed that the states and control input of the target at the initial movement obey Gauss distribution. The mean and covariance of the control input are denoted by $u_{T0}$ and $\sigma_{u_T}^2$, respectively, and the mean and covariance matrix of the states are denoted by $X_T(0)$ and $Q_{X_T}$, respectively. Let $u_T(k) = u_{T0}$, then the mean value of the target movement states at $t = t_e$ can be predicted by the iteration calculation of Equation (11). In addition, the covariance matrix of the target movement state at $t = t_e$ can be obtained by the iterative calculation of the following equation:

$$Q_{X_T}(k+1) = A_k Q_{X_T}(k) A_k^{\mathrm{T}} + B_k \sigma_{u_T}^2 B_k^T \tag{12}$$

where

$$A_k = \left. \frac{\partial f_T}{\partial X_T} \right|_{X_T = X_T(k)}, \ B_k = \left. \frac{\partial f_T}{\partial u_T} \right|_{X_T = X_T(k)} \tag{13}$$

In particular, if the target movement model is linear, i.e.,

$$\dot{X}_T = A_T X_T + B_T u_T \tag{14}$$

then, under the initial conditions, the mean and covariance matrix of the target movement state at $t = t_e$ are

$$
\begin{aligned}
&X_T(t_e) = \Phi_T(t_e, t_0) X_T(0) + \int_{t_0}^{t_e} \Phi_T(t_e, \tau) B_T u_{T0} d\tau \\
&Q_{X_T}(t_e) = (\Phi_T(t_e, t_0))^{\mathrm{T}} Q_{X_T}(0) \Phi_T(t_e, t_0) + \int_{t_0}^{t_e} \Phi_T(t_e, \tau) B_T u_{T0} d\tau \cdot \sigma_{u_T}^2 \cdot \left( \int_{t_0}^{t_e} \Phi_T(t_e, \tau) B_T u_{T0} d\tau \right)^{\mathrm{T}}
\end{aligned} \tag{15}
$$

where $A_T$ and $B_T$ are the system matrix and input matrix with proper dimensions, and $\Phi_T(t_e, t_0)$ is the state transition matrix of system (14).

By the iteration calculation of Equation (15) or Equations (11) and (12), the mean and covariance matrix of the target movement state at $t_e$ can be obtained, and then the mean value and covariance matrix of the target location at $t_e$ can be calculated, which are denoted by $x_T(t_e)$ and $Q_{x_T}$, respectively. Then, the predicted interception area can be described as

$$\mathcal{R} = \left\{ x \mid x \sim \mathcal{N}\left( x_T(t_e), Q_{x_T} \right) \right\} \tag{16}$$

*2.3. Problem Formulation of Simultaneous Cooperative Interception*

In this subsection, some assumptions are given and then the cooperative interception problem is formulated based on area coverage.

**Assumption 4:** *The distances between the interceptors are rather small compared with that between the interceptors and the target, thus the line-of-sight angles can be approximated to be the same and the movement models of all interceptors can be described in the same line-of-sight frame.*

**Assumption 5:** *The predicted interception moment between an interceptor and the target is assumed to be known, and all the interceptors are assumed to encounter the target at the same time.*

**Assumption 6:** *The effects of target maneuver and movement information errors on the predicted interception moment can be neglected.*

Now, a performance index function of the handover of cooperative interception will be constructed based on area coverage. First, the definition of the target-interceptor impact plane is presented.

**Definition 5:** *Target–interceptor impact plane: a plane passing through the TPZC of an interceptor and perpendicular to the line of sight is called the target–interceptor impact plane.*

Since the acceleration of an interceptor is perpendicular to the line of sight, the predicted interception area can be projected onto the target–interceptor impact plane. Based on the above assumptions and definitions, the cooperative interception engagement of $N$ interceptors can be shown in Figure 2, where the plane EFGH is the target–interceptor impact plane. $I_j$ and $T$ denote the $j$-th interceptor and target, and $V_j$ and $V_T$ are the velocities of $I_j$ and the target, respectively. $P_j$ denotes the TPZC of interceptor $I_j$. $\overline{\mathcal{R}}$ is the projection of the predicted interception area onto the plane EFGH and $P_T$ is the center of $\overline{\mathcal{R}}$. The calculation of $\overline{\mathcal{R}}$ can be found in the Appendix A.

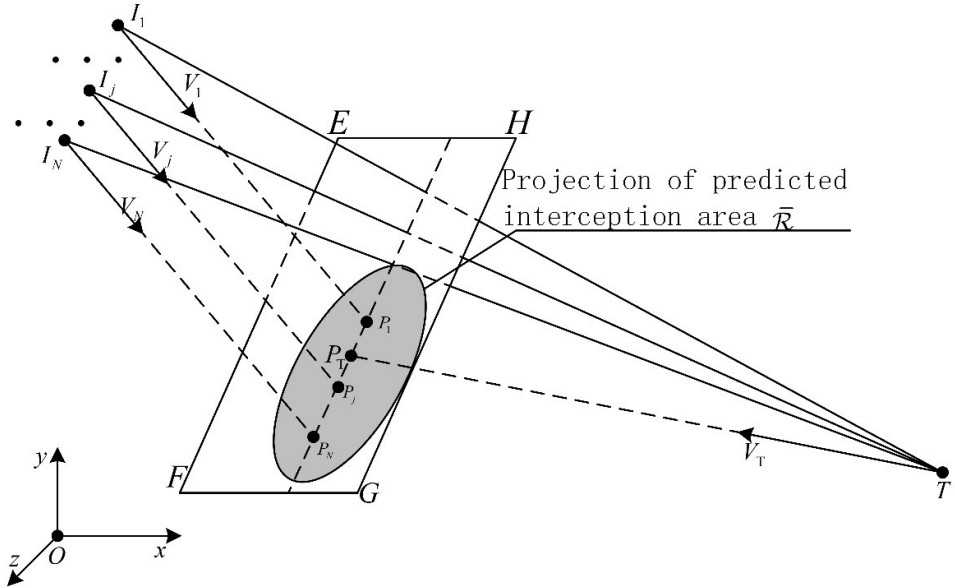

**Figure 2.** Cooperative interception engagement geometry between multiple interceptors and a target.

Let $\boldsymbol{p}_j = [p_j^y, \ p_j^z]^\mathrm{T}$ be the coordinates of the TPZC of $I_j$ at the handover moment. For a point $\boldsymbol{\xi}$ within the area $\overline{\mathcal{R}}$, the miss distance of $\boldsymbol{\xi}$ with respect to interceptor $I_j$ is

$$Z\left(\boldsymbol{\xi}, \boldsymbol{p}_j\right) = \left\| \boldsymbol{p}_j - \boldsymbol{\xi} \right\| \tag{17}$$

Considering that $\boldsymbol{\xi}$ has a miss distance with respect to every interceptor $I_j (j = 1, \cdots, N)$, choose the minimum one as the miss distance of $\boldsymbol{\xi}$, i.e.,

$$Z(\boldsymbol{\xi}, \boldsymbol{p}_1, \cdots, \boldsymbol{p}_N) = \min_{j=1,\cdots,N} \left\| \boldsymbol{p}_j - \boldsymbol{\xi} \right\| \tag{18}$$

Since the target may eventually appear at any point in the predicted interception area, let $\phi(\boldsymbol{\xi})$ be the probability density function of $\boldsymbol{\xi}$, which describes the probability that the target appears at point $\boldsymbol{\xi}$. Thus, the expectation of the miss distance with respect to the target can be expressed as

$$\mathbb{E}[Z] = \int_{\overline{\mathcal{R}}} \left( \min_{j=1,\cdots,N} \| \boldsymbol{p}_j - \boldsymbol{\xi} \| \right) \phi(\boldsymbol{\xi}) d\boldsymbol{\xi} \tag{19}$$

Next, the probability of a successful handover for simultaneous cooperative interception is presented. Let $t_f$ be the total time of the terminal guidance, then the maximum distance that an interceptor can maneuver in the terminal guidance is

$$d_0 = \frac{1}{2} a_{\max} t_f^2 \tag{20}$$

According to Equation (9), the reachable area of interceptor $I_j$ in the plane EFGH can be expressed as

$$\overline{\mathcal{M}}_j = \left\{ (y, z) \Big| \left| y - p_j^y \right| \leq d_0, \ \left| z - p_j^z \right| \leq d_0 \right\} \tag{21}$$

Let $f_j(\boldsymbol{p}_j, \boldsymbol{\xi})$ be an indicator function describing whether or not a point $\boldsymbol{\xi}$ is within the reachable area of $I_j$, which is expressed by

$$f_j(\boldsymbol{p}_j, \boldsymbol{\xi}) = \begin{cases} 1, & \left| y - p_j^y \right| \leq d_0, \ \left| z - p_j^z \right| \leq d_0 \\ 0, & \text{else} \end{cases} \tag{22}$$

where $\xi_y$ and $\xi_z$ are the coordinate components of $\boldsymbol{\xi}$, i.e., $\boldsymbol{\xi} = \left[ \xi_y, \xi_z \right]^{\mathrm{T}}$. Let $F_c$ be the function describing whether or not a point $\boldsymbol{\xi}$ is within the reachable areas of $N$ interceptors, which can be described as

$$F_c(\boldsymbol{\xi}, \boldsymbol{p}_1, \cdots, \boldsymbol{p}_N) = 1 - \prod_{j=1}^{N} \left( 1 - f_j(\boldsymbol{p}_j, \boldsymbol{\xi}) \right) \tag{23}$$

If $F_c = 1$, there exists an interceptor such that $f_j(\boldsymbol{p}_j, \boldsymbol{\xi}) = 1$, i.e., the point $\boldsymbol{\xi}$ is within the reachable area of $I_j$. Let $P_{rh}$ be the probability of a successful handover for $N$ interceptors, then $P_{rh}$ can be expressed by

$$P_{rh} = \int_{\mathcal{R}} \left( 1 - \prod_{j=1}^{N} \left( 1 - f_j(\boldsymbol{p}_j, \boldsymbol{\xi}) \right) \right) \phi(\boldsymbol{\xi}) \mathrm{d}\boldsymbol{\xi} \tag{24}$$

At the handover moment, the smaller the miss distance, the more conducive the interception of the target in the terminal guidance [7]. Thus, in order to increase the probability of hitting the target, the expectation of the miss distance should be taken as the performance index to optimize the TPZCs of $N$ interceptors. In addition, in order to guarantee the probability of a successful handover, the number of interceptors should reach a certain value. Let $Y_{\min}$ be the required minimum value of $P_{rh}$, then considering both the miss distance and probability of a successful handover, the simultaneous cooperative interception design problem can be described as follows:

$$\begin{aligned} \min_{\boldsymbol{p}_1, \cdots, \boldsymbol{p}_N} J &= \int_{\mathcal{R}} \left( \min_{j=1, \cdots, N} \left\| \boldsymbol{p}_j - \boldsymbol{\xi} \right\| \right) \phi(\boldsymbol{\xi}) \mathrm{d}\boldsymbol{\xi} \\ \text{s.t. } & P_{rh}(N, \boldsymbol{p}_1, \cdots, \boldsymbol{p}_N) \geq Y_{\min} \end{aligned} \tag{25}$$

By solving the optimization problem of Equation (25), the minimum number of interceptors and the corresponding TPZC of each interceptor at the handover moment can be obtained, which not only satisfies the demand for the probability of a successful handover but also helps to intercept the target in the terminal guidance.

### 3. Simultaneous Cooperative Interception Design Based on Area Coverage Optimization

In this section, we will first propose an approach to solve the optimization problem which is formulated in Section 2. For the simultaneous cooperative interception problem described by Equation (25), the number of interceptors is an integer, which cannot be obtained by the continuous algorithm and is mainly related to the probability of a successful handover. Then, the simultaneous cooperative interception problem can be solved in two

steps. First, optimize the TPZC of each interceptor to minimize the miss distance for a given number $N$, and then search for the minimum number of interceptors that satisfies the constraints for the probability of a successful handover through certain iterative rules. In this section, an area coverage optimization algorithm to calculate the TPZCs of interceptors is presented, and then a solution for the simultaneous cooperative interception problem is proposed based on the area coverage optimization algorithm.

### 3.1. Area Coverage Optimization Algorithm for Simultaneous Cooperative Interception

For the performance index function (19), $\min\limits_{j=1,\cdots,N}\left\|\boldsymbol{p}_j - \boldsymbol{\xi}\right\|$ means the division of the integral area $\overline{\mathcal{R}}$, and the division results are $N$ irregular subareas, which makes the integral calculation more difficult. Considering that $\phi(\boldsymbol{\xi})$ is the probability density function of the area $\overline{\mathcal{R}}$, the commonly used approach for calculating this kind of integral function is a stochastic approximation, i.e., approximating the integral function by Monte Carlo sampling [23]. Thus, an algorithm based on area division and stochastic approximation is presented in this section.

For a point $\boldsymbol{\xi}$ in the area $\overline{\mathcal{R}}$, based on the definition of miss distance described by Equation (18), the area $\overline{\mathcal{R}}$ can be divided into $N$ subareas, which are denoted by $\overline{\mathcal{R}}_j (j = 1, \cdots, N)$. The expression of $\overline{\mathcal{R}}_j$ is

$$\overline{\mathcal{R}}_j = \left\{ \boldsymbol{\xi} \in \overline{\mathcal{R}} \,\middle|\, \left\|\boldsymbol{\xi} - \boldsymbol{p}_j\right\| \leq \left\|\boldsymbol{\xi} - \boldsymbol{p}_i\right\|, \, \forall i \neq j \right\} \tag{26}$$

For any point $\boldsymbol{\xi}$ in the subarea $\overline{\mathcal{R}}_j$, the miss distance of $\boldsymbol{\xi}$ with respect to interceptor $I_j$ is smaller than that of any other interceptor. Thus, the miss distance of $\boldsymbol{\xi}$ can be rewritten as

$$Z(\boldsymbol{\xi}, \boldsymbol{p}_1, \cdots, \boldsymbol{p}_N) = \left\|\boldsymbol{p}_j - \boldsymbol{\xi}\right\|, \text{ if } \boldsymbol{\xi} \in \overline{\mathcal{R}}_j \tag{27}$$

Let $\boldsymbol{\xi} = [\xi_y, \xi_z]^{\mathrm{T}}$, and based on Equation (27), the expectation of the miss distance can be re-expressed as

$$\begin{aligned} J &= \sum_{j=1}^{N} \int_{\overline{\mathcal{R}}_j} \left\|\boldsymbol{p}_j - \boldsymbol{\xi}\right\| \phi(\boldsymbol{\xi}) \mathrm{d}\boldsymbol{\xi} \\ &= \sum_{j=1}^{N} \int_{\overline{\mathcal{R}}_j} \sqrt{\left|p_j^y - \xi_y\right|^2 + \left|p_j^z - \xi_z\right|^2} \, \phi(\boldsymbol{\xi}) \mathrm{d}\boldsymbol{\xi} \end{aligned} \tag{28}$$

Based on Assumption 3, the cooperative interception performance index function can be redescribed as

$$J = \sum_{j=1}^{N} \int_{\overline{\mathcal{R}}_j} \left( \left|p_j^y - \xi_y\right| + \left|p_j^z - \xi_z\right| \right) \phi(\boldsymbol{\xi}) \mathrm{d}\boldsymbol{\xi} \tag{29}$$

Now, we will present an algorithm to solve the optimization problem. For the performance index function in Equation (29), let

$$J_y = \sum_{j=1}^{N} \int_{\overline{\mathcal{R}}_j} \left|p_j^y - \xi_y\right| \phi(\boldsymbol{\xi}) \mathrm{d}\boldsymbol{\xi} \tag{30}$$

$$J_z = \sum_{j=1}^{N} \int_{\overline{\mathcal{R}}_j} \left|p_j^z - \xi_z\right| \phi(\boldsymbol{\xi}) \mathrm{d}\boldsymbol{\xi} \tag{31}$$

then we have $J = J_y + J_z$ and the optimization of Equation (29) can be solved by the optimization of $J_y$ and $J_z$, respectively.

Now, we will take $J_y$ as an example and outline the process of solving the cooperative interception problem. First, in order to calculate the integral on the region $\overline{\mathcal{R}}_j$, the Monte Carlo approach is used to obtain the samples of $\xi$ according to the probability density function $\phi(\xi)$, which are denoted by $\xi_1, \cdots, \xi_{N_\xi}$, where $N_\xi$ is the sample size. Based on the area division approach, the samples in the subarea $\overline{\mathcal{R}}_j$ are denoted by $Q_j = \left\{ \xi_{j1}, \cdots, \xi_{jn_j} \right\}$, where $n_j$ is the total number of samples in $Q_j$. Based on the sampling, the function $J_y$ can be approximated by

$$J_y = \frac{1}{N_\xi} \sum_{j=1}^{N} \sum_{k=1}^{n_j} \left| p_j^y - \xi_{jk}^y \right| \tag{32}$$

where $\xi_{jk}^y$ is the coordinate of $\xi_{jk}$ in the $Oy$ direction.

Since $p_j \in \overline{\mathcal{R}}_j$, $p_j^y$ should satisfy some constraints. Assuming that the maximum and minimum values of $p_j^y$ are $y_{j\min}$ and $y_{j\max}$, respectively, i.e.,

$$y_{j\min} \leq p_j^y \leq y_{j\max} \tag{33}$$

Next, we will solve the minimum value of $J_y$ with respect to $p_j^y$ under the constraints of Equation (33). Sort the samples in $Q_j$ in ascending order of $\xi_{jk}^y$, which are denoted by $\xi_{j1}^y \leq \xi_{j2}^y \leq \cdots \leq \xi_{jn_j}^y$, then the derivative of $J_y$ with respect to $p_j^y$ can be expressed by

$$\frac{\partial J_y}{\partial p_j^y} = \begin{cases} -n_j, \ p_j^y \leq \xi_{j1}^j \\ -(n_j - 2), \ \xi_{j2}^j \leq p_j^y \leq \xi_{j3}^j \\ \vdots \\ -1, \ \xi_{jk_1}^j \leq p_j^y \leq \xi_{j(k_1+1)}^j \\ 1, \ \xi_{j(k_1+1)}^j \leq p_j^y \leq \xi_{j(k_1+2)}^j \\ \vdots \\ n_j, \ \xi_{jn_j}^y \leq p_j^y \end{cases} , \ n_j \text{ is an odd number} \tag{34}$$

$$\frac{\partial J_y}{\partial p_j^y} = \begin{cases} -n_j, \ p_j^y \leq \xi_{j1}^j \\ -(n_j - 2), \ \xi_{j2}^j \leq p_j^y \leq \xi_{j3}^j \\ \vdots \\ 0, \ \xi_{jk_1}^j \leq p_j^y \leq \xi_{j(k_1+1)}^j \\ \vdots \\ n_j, \ \xi_{jn_j}^y \leq p_j^y \end{cases} , \ n_j \text{ is an even number} \tag{35}$$

where $k_1 = \frac{n_j - 1}{2}$ if $n_j$ is an odd number and $k_1 = \frac{n_j}{2}$ if $n_j$ is an even number.

If $n_j$ is an odd number, $\frac{\partial J_y}{\partial p_j^y} < 0$ when $p_j^y \leq \xi_{j(k_1+1)}^y$, and $\frac{\partial J_y}{\partial p_j^y} > 0$ when $p_j^y \geq \xi_{j(k_1+1)}^y$, thus the extreme point is $\hat{p}_j^y = \xi_{j(k_1+1)}^y$. If $n_j$ is an even number, $\frac{\partial J_y}{\partial p_j^y} = 0$ when $\xi_{jk_1}^y \leq p_j^y \leq \xi_{j(k_1+1)}^y$, then the extreme point can be chosen to be $\xi_{jk_1}^y$. Then, the extreme point of $Jy$ with respect to $p_j^y$ is

$$\hat{p}_j^y = \begin{cases} \xi_{j(k_1+1)}^y, & \text{if } n_j \text{ is an odd number} \\ \xi_{jk_1}^y, & \text{if } n_j \text{ is an even number} \end{cases} \tag{36}$$

Since $\frac{\partial J_y}{\partial p_j^y} < 0$ when $p_j^y \leq \hat{p}_j^y$ and $\frac{\partial J_y}{\partial p_j^y} \geq 0$ when $p_j^y \geq \hat{p}_j^y$, $\hat{p}_j^y$ is a minimum point. Combined with the constraints in Equation (33), the minimum point of $Jy$ with respect to $p_j^y$ is

$$\widetilde{p}_j^y = \begin{cases} y_{j\min}, & \text{if } \hat{p}_j^y \leq y_{j\min} \\ \hat{p}_j^y, & \text{if } y_{j\min} \leq \hat{p}_j^y \leq y_{j\max} \\ y_{j\max}, & \text{if } y_{j\max} \leq \hat{p}_j^y \end{cases} \tag{37}$$

Similarly, sort the samples in $Q_j$ in ascending order of $\xi_{jk}^z$, which are denoted by $\xi_{j1}^z \leq \xi_{j2}^z \leq \cdots \leq \xi_{jn_j}^z$, then the minimum point of $J_z$ with respect to $p_j^z$ is

$$\widetilde{p}_j^z = \begin{cases} z_{j\min}, & \text{if } \hat{p}_j^z \leq z_{j\min} \\ \hat{p}_j^z, & \text{if } z_{j\min} \leq \hat{p}_j^z \leq z_{j\max} \\ z_{j\max}, & \text{if } z_{j\max} \leq \hat{p}_j^z \end{cases} \tag{38}$$

where $z_{j\min}$ and $z_{j\max}$ are the minimum and maximum values of $p_j^z$, respectively, and the expression of $\hat{p}_j^z$ is

$$\hat{p}_j^z = \begin{cases} \xi_{j(k_1+1)}^z, & \text{if } n_j \text{ is an odd number} \\ \xi_{jk_1}^z, & \text{if } n_j \text{ is an even number} \end{cases} \tag{39}$$

After a division of the predicted interception area, the optimal solution of $J$ can be obtained according to Equations (37) and (38), which is denoted by $\widetilde{p}_j = [\widetilde{p}_j^y, \widetilde{p}_j^z]$ $(j = 1, \cdots, N)$. Then, divide the predicted interception area again based on $\widetilde{p}_j$, and solve the optimal solution of $J$, until the global optimum is achieved. Thus, the area coverage optimization algorithm for simultaneous cooperative interception can be summarized as follows.

---

**Algorithm 1** Area coverage optimization algorithm

---

1. Step 1: sample the predicted interception area $\overline{\mathcal{R}}$ based on the probability density function, and the samples are denoted by $\xi_1, \cdots, \xi_{N_\xi}$;
2. Step 2: group the samples according to Equation (26);
3. Step 3: solve the minimum point of the cooperative interception performance index with respect to $p_j^y$ and $p_j^z$ according to Equations (37) and (39);
4. Step 4: return to Step 2.

---

**Remark 1.** *For the area coverage optimization algorithm, the convergence is proved in the Appendices A and B. When the number of iterations tends to infinity, the performance index will converge to the optimal solution. In order to meet the real-time demands of the cooperative interception, it is necessary to set the terminating condition of the algorithm, and the iteration can be stopped when the accuracy of the solution meets the requirements. The terminating condition can be chosen to be $\sum_{j=1}^{N} \left\| \widetilde{p}_j(i+1) - \widetilde{p}_j(i) \right\| \leq v$, where $i$ is the number of iterations and $v$ is the requirements of solving the precision.*

*3.2. Simultaneous Cooperative Interception Design*

Now, an approach to solving the cooperative interception design problem is proposed based on the area coverage optimization algorithm. According to Monte Carlo sampling, *Prh* can be approximated by

$$P_{rh} = \frac{1}{N_\xi} \sum_{i=1}^{N_\xi} \left(1 - \prod_{j=1}^{N} \left(1 - f_j(\boldsymbol{p}_j, \boldsymbol{\xi}_i)\right)\right) \tag{40}$$

For the simultaneous cooperative interception, after obtaining the TPZCs of $N$ interceptors, if $P_{rh} \geq Y_{min}$, the number of interceptors can be reduced to save combat costs. However, if $P_{rh} < Y_{min}$, the number of interceptors should be increased to meet the interception requirements. Thus, the minimum value of $N$ can be obtained by successively increasing or reducing the number of interceptors. In order to increase the speed of the calculation, the algorithm of dichotomy is used, and the iteration process can be summarized as follows:

1.  Step 1: set the initial number of interceptors to be $N_0$ and let $i = k = l = 0$;
2.  Step 2: let $N = Ni$, $i = i + 1$;
3.  Step 3: solve the TPZCs of $Ni$ interceptors based on the area coverage optimization algorithm;
4.  Step 4: calculate the probability of a successful handover for $N_i$ interceptors according to Equation (40);
5.  Step 5: if $P_{rh} \geq Y_{min}$ and $N_{i-1} - l \neq 1$, then $k = Ni\text{-}1$, $N_i = \left\lfloor \frac{k+l}{2} \right\rfloor$ and return to step 2; if $P_{rh} \geq Y_{min}$ and $N_{i-1} - l = 1$, then $N_i^* = N_{i-1}$ and stop iterating; if $P_{rh} < Y_{min}$ and $k = 0$, then $Ni = 2\, l$ and return to step 2; if $P_{rh} < Y_{min}$ and $k \neq 0$, then $N_i = \left\lfloor \frac{k+l}{2} \right\rfloor$ and return to step 2.

During the iteration process, $N_0$ is an initial value, and $k$ and $l$ are intermediate variables that denote the number of interceptors that satisfy $P_{rh} \geq Y_{min}$ and $P_{rh} < Y_{min}$, respectively. $\lfloor x \rfloor$ denotes the maximum integer that is not larger than $x$.

**Remark 2.** *The critical steps in the iteration process are steps 3 and step 4, i.e., searching for the optimal TPZCs of the interceptors based on the area coverage optimization algorithm and calculating the probability of a successful handover according to Equation (40). In step 5, whether the iteration process stops and the number of interceptors needed for the next cycle of iterations are determined. The design process is illustrated in Figure 3.*

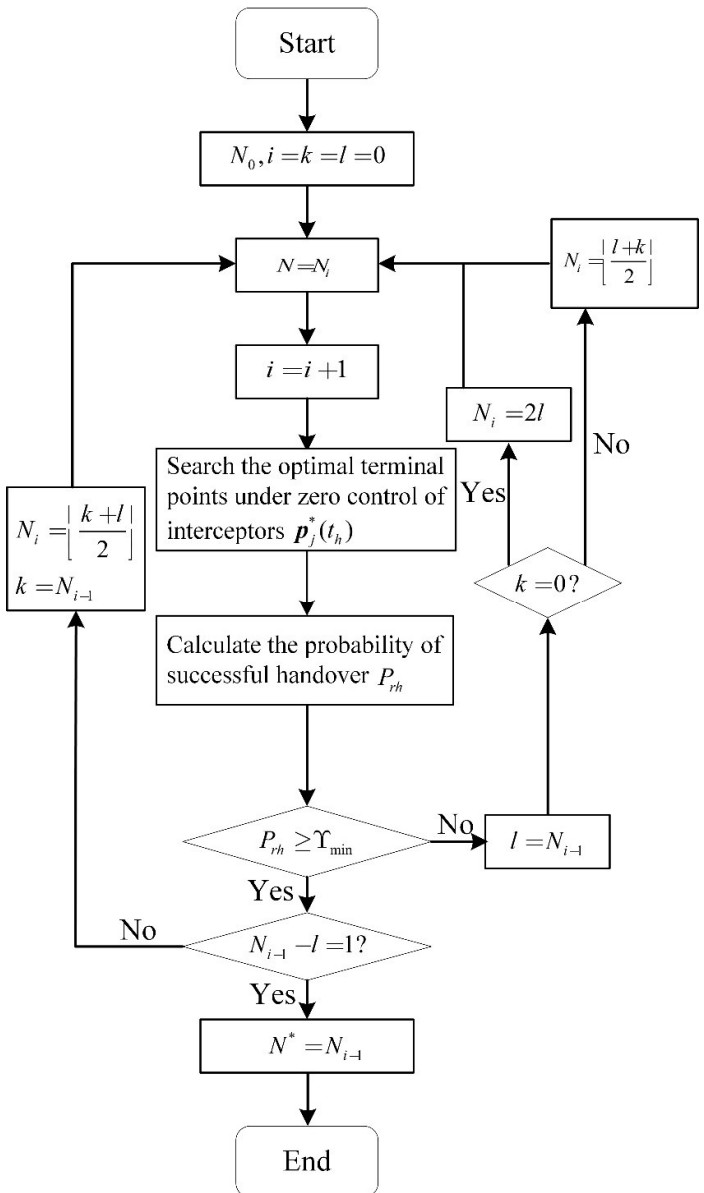

**Figure 3.** The flow diagram of simultaneous cooperative interception design.

## 4. Simulation Experiments and Analysis

In this section, some simulation experiments will be carried out to illustrate the effectiveness of the proposed cooperative interception design approach. Consider the interception of a high-speed invading target, such as a ballistic missile or hypersonic flight vehicle in the near space. The target movement information is provided by the ground-based tracking system or space-based tracking system. The error characteristics of the tracking system are set up as follows. Let $[\Delta x_T, \Delta y_T, \Delta z_T]^T$, $[\Delta v_{xT}, \Delta v_{yT}, \Delta v_{zT}]^T$ and $[\Delta a_{xT}, \Delta a_{yT}, \Delta a_{zT}]^T$ be the errors of position, velocity, and acceleration of the target in the inertial frame, respectively, which satisfy

$$
\begin{aligned}
&|\Delta x_T| \leq 1\text{Km}, \ |\Delta y_T| \leq 1\text{Km}, \ |\Delta z_T| \leq 1\text{Km} \\
&|\Delta v_{xT}| \leq 100\,\text{m/s}, \ |\Delta v_{yT}| \leq 100\,\text{m/s}, \ |\Delta v_{zT}| \leq 100\,\text{m/s} \\
&|\Delta a_{xT}| \leq 100\,\text{m/s}^2, \ |\Delta a_{yT}| \leq 100\,\text{m/s}^2, \ |\Delta a_{zT}| \leq 100\,\text{m/s}^2
\end{aligned}
\tag{41}
$$

The elevation and azimuth angles of the initial line-of-sight frame with respect to the inertial frame are set to be $-5°$ and $0°$. Then according to the coordinate transformation,

the target movement information errors along the $Oy$ and $Oz$ axes in the initial line-of-sight frame satisfy

$$
\begin{aligned}
&|\Delta y'_T| \leq 1\text{Km}, \; |\Delta z'_T| \leq 1\text{Km} \\
&|\Delta v'_{yT}| \leq 100 \text{ m/s}, \; |\Delta v'_{yT}| \leq 100 \text{ m/s} \\
&|\Delta a'_{yT}| \leq 100 \text{ m/s}^2, \; |\Delta a'_{zT}| \leq 100 \text{ m/s}^2
\end{aligned}
\tag{42}
$$

The total time of the terminal guidance is set to be $t_f$ = 10 s, then the projection of the predicted interception area onto the target–interceptor impact plane can be expressed as

$$
\overline{\mathcal{R}} = \{(y,z)||y| \leq 2708.3 \text{ m}, \; |z| \leq 2500 \text{ m}\}
\tag{43}
$$

Thus, the variance of the target movement information errors can be set as

$$
\sigma_y^2 = \left(\frac{2708.3}{3}\right)^2, \; \sigma_z^2 = \left(\frac{2500}{3}\right)^2
$$

Assuming that the distribution of target movement information errors along the $Oy$ axis and $Oz$ axis are independent of each other, then the probability density function of $\overline{\mathcal{R}}$ is

$$
\phi(\xi) = \frac{1}{2\pi\sigma_y\sigma_z}\exp\left\{-\frac{1}{2}\left(\frac{\xi_y^2}{\sigma_y^2} + \frac{\xi_z^2}{\sigma_z^2}\right)\right\}
\tag{44}
$$

The maximum acceleration of an interceptor in the terminal guidance is set at 30 m/s$^2$, then the maximum maneuverable distance is $d_0$ = 1500 m. The lower bound of successful handover probability is chosen to be $Y_{\min}$ = 95%.

Next, considering both the miss distance and the demands of a successful handover probability, the number of interceptors and the corresponding TPZC of each interceptor at the handover moment are designed. The initial number of interceptors is set to be $N_0$ = 2, then according to the design process of cooperative interception, after a certain number of iterations, it can be ascertained that at least four interceptors are needed to satisfy the demands of a successful handover probability. For the cooperative interception of the four interceptors under the iterative calculation of the area coverage optimization algorithm, the TPZCs of the four interceptors, the expectation of miss distance, and the probability of a successful handover are shown in Figure 4. In Figure 4a we can see that the optimal TPZCs of the four interceptors with respect to the center of the predicted interception area are (−1450 m, −1250 m), (−1450 m, 1250 m), (1450 m, −1250 m), and (1450 m, 1250 m), respectively.

According to the results of the calculation, we can see that under the optimal distribution of the TPZCs, the successful handover probability of the four interceptors is 99.5%. However, if all the interceptors are aiming at the center of the predicted interception area, the successful handover probability is 83%.

In the office computer environment, the computing time of the proposed algorithm is about 0.67s, which can meet the real-time requirements of operations.

In the following sections, some interception experiments are carried out to validate the effectiveness of the designed results. In the mid-course guidance phase, every interceptor flies to its TPZC under the proportional guidance law, and in the terminal guidance phase, every interceptor flies to the target. The initial movement states of the four interceptors and the target are shown in Table 1. Four different interception cases are considered here, and the target movement information errors in each case are all set at the maximum, which are shown in Table 2. The cooperative interception trajectories in different cases are presented in Figure 5, and the terminal miss distance of each interceptor is given in Table 3. From the results of the cooperative interception, it can be seen from the distribution of the TPZCs at the handover moment, that at least one interceptor is successfully handing over and hitting the target with a rather small miss distance in the terminal guidance. When the interceptors do not cooperate, i.e., every interceptor flies to the center of the predicted interception

area under proportional guidance law in the mid-course guidance phase. The interception trajectories and terminal distance of each interceptor are presented in Figure 6 and Table 4, respectively. From the results in Table 4, it can be seen that all interceptors fail to hit the target with a large miss distance.

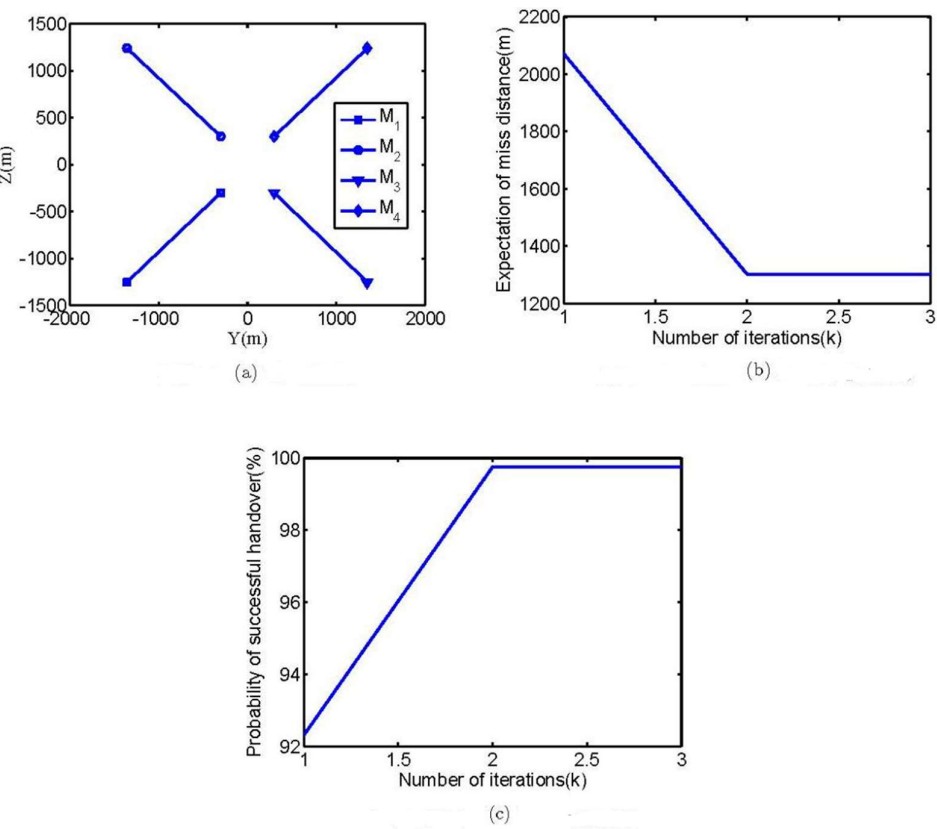

**Figure 4.** Simultaneous cooperative interception design of four interceptors. (**a**) TPZCs of four interceptors; (**b**) Expectation of zero effort miss distance; (**c**) Probability of successful handover.

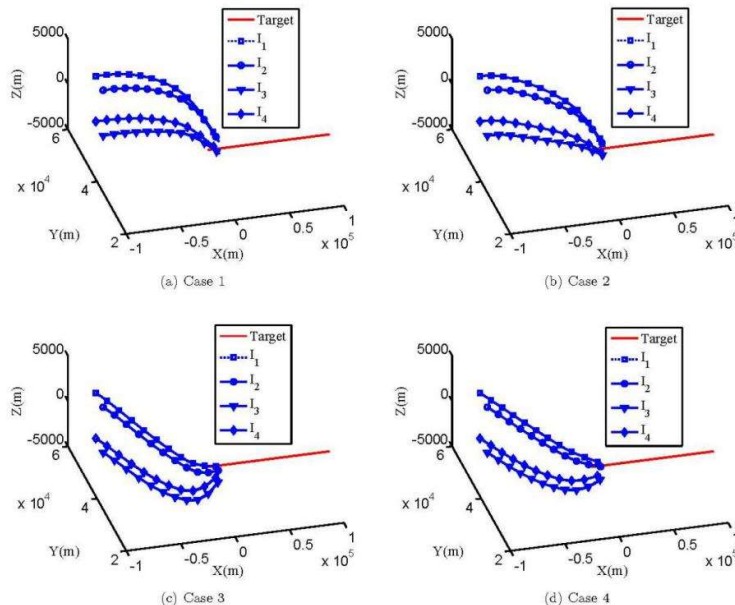

**Figure 5.** Cooperative interception trajectories in different cases. (**a**) Case 1; (**b**) Case 2; (**c**) Case 3; (**d**) Case 4.

**Table 1.** Initial conditions of interceptors and targets.

| Interceptor/Target | Position (km) | Velocity (m/s) |
|:---:|:---:|:---:|
| $I_1$ | $(-87.476, 50.208, 3.5)$ | $(2097.8, -312.8, 100)$ |
| $I_2$ | $(-87.948, 44.812, 3.5)$ | $(2097.8, -312.8, 100)$ |
| $I_3$ | $(-87.948, 44.812, -1.5)$ | $(2097.8, -312.8, 100)$ |
| $I_4$ | $(-87.476, 50.208, -1.5)$ | $(2097.8, -312.8, 100)$ |
| Target | $(100, 30, 0)$ | $(-2720, -0, 0)$ |

**Table 2.** Errors of target movement information in different interception cases.

| Interception Cases | Error of Target Position (m) | Error of Target Velocity (m/s) |
|:---:|:---:|:---:|
| Case 1 | $(1000, 1000, 1000)$ | $(100, 100, 100)$ |
| Case 2 | $(-1000, -1000, 1000)$ | $(-100, -100, 100)$ |
| Case 3 | $(-1000, -1000, -1000)$ | $(-100, -100, -100)$ |
| Case 4 | $(1000, 1000, -1000)$ | $(100, 100, -100)$ |

**Table 3.** Miss distances of cooperative interception.

| Interception Cases | $I_1$ | $I_2$ | $I_3$ | $I_4$ |
|:---:|:---:|:---:|:---:|:---:|
| Case 1 | 2673.2 m | 1534.9 m | 0.24 m | 2029.2 m |
| Case 2 | 1781.9 m | 2394.5 m | 1828.8 m | 0.27 m |
| Case 3 | 0.22 m | 1852.7 m | 2383.8 m | 1712.6 m |
| Case 4 | 2012.9 m | 0.19 m | 1511.7 m | 2625.4 m |

**Table 4.** Miss distances of non-cooperative interception.

| Interception Cases | $I_1$ | $I_2$ | $I_3$ | $I_4$ |
|:---:|:---:|:---:|:---:|:---:|
| Case 1 | 1665.2 m | 1251.5 m | 1168.7 m | 1427.3 m |
| Case 2 | 1539.1 m | 1273.5 m | 1195.4 m | 1263.1 m |
| Case 3 | 1428.2 m | 1280.2 m | 1214.4 m | 1373.9 m |
| Case 4 | 1569.7 m | 1256.2 m | 1190.7 m | 1521.1 m |

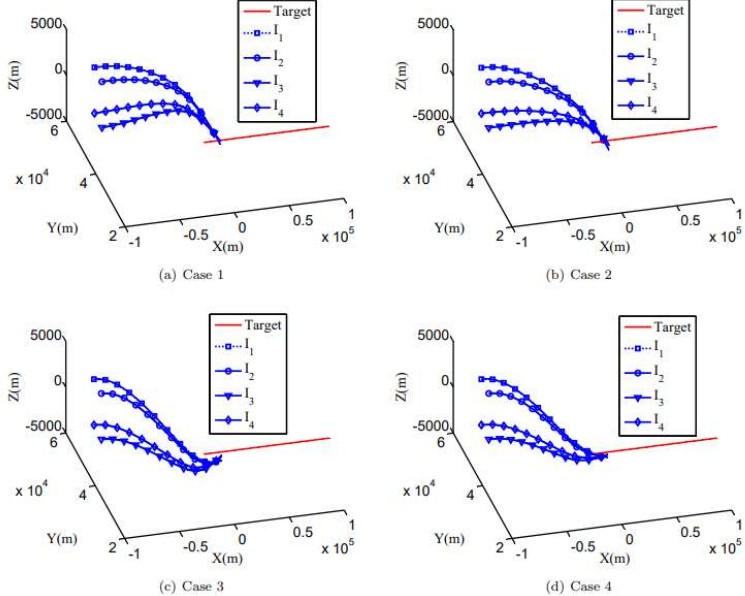

**Figure 6.** Non-cooperative interception trajectories in different cases. (**a**) Case 1; (**b**) Case 2; (**c**) Case 3; (**d**) Case 4.

The results of the simulation experiments show that in non-cooperative mode, the interceptors move close to each other in the lateral direction, and then all the handover

errors of the four interceptors are larger than the maximum maneuverable distance, which results in the failure of the interception. In cooperative mode, every point in the predicted interception area is within the reachable area of at least one interceptor by the spatial distribution of the TPZCs, which guarantees the successful handover of at least one interceptor and hitting the target with a small miss distance in the terminal guidance.

## 5. Conclusions

This paper deals with the problem of simultaneous cooperative interception using multiple interceptors. Considering the target maneuver and movement information errors, the simultaneous cooperative interception design problem is formulated based on area coverage. An area coverage optimization algorithm is presented, which is based on the cooperative interception design approach. The number of interceptors and the corresponding spatial distribution are designed based on the proposed approach, which guarantees that there is at least one interceptor capable of hitting the target in the terminal guidance. The effectiveness of the proposed approach is validated by the simulation results.

Since the predicted interception area is changing with the variation in the target movement information errors in practical engagement, the spatial distributions of the interceptors should be adjusted according to the changes in the predicted interception area. Thus, future work in this direction could be focused on the problem of cooperative mid-course guidance law design.

**Author Contributions:** Conceptualization, L.W. and Y.Y.; methodology, L.W. and F.H.; validation, L.W. and K.L.; writing—original draft preparation, L.W.; writing—review and editing, K.L.; supervision, Y.Y. All authors have read and agreed to the published version of the manuscript.

**Funding:** This research was funded by NSFC, grant number U2141229.

**Institutional Review Board Statement:** Not applicable.

**Informed Consent Statement:** Not applicable.

**Data Availability Statement:** The data that support the findings of this study are available from the corresponding author, Kai Liu, upon reasonable request.

**Conflicts of Interest:** The authors declare no conflict of interest.

## Appendix A. Projection of the Predicted Interception Area

Before calculating the projection of the predicted interception area, the following definitions are given.

**Definition A1:** *Inertial frame* $Ox_Iy_Iz_I$*: the origin* $O$ *is chosen to be the mass center of the interceptor at the launch moment,* $Ox_I$ *points to the launch direction,* $Oy_I$ *lies in the vertical plane containing the* $Ox_I$ *axis and perpendicular to the* $Ox_I$ *axis,* $Oz_I$ *is determined by the right-hand rule.*

The relationship between the line-of-sight frame and inertial frame can be described by two angles, which are defined as follows.

**Definition A2:** *Elevation angle of the line of sight* $q_\varepsilon$*: the angle between the line of sight and the* $Ox_Iz_I$ *plane. If the line of sight is above the* $Ox_Iz_I$ *plane, then* $q_\varepsilon$ *is positive and vice versa.*

**Definition A3:** *Azimuth angle of the line of sight* $q_\beta$*>: the angle between the* $Ox_I$ *axis and the projection of the line of sight in the* $Oxz$ *plane. Looking towards the direction of the* $Ox_I$ *axis, if the* $Ox$ *axis rotates to the projection anticlockwise, then* $q_\beta$ *is positive and vice versa.*

According to the definitions of $q_\varepsilon$ and $q_\beta$, the line-of-sight frame can be obtained by two rotations of the inertial frame. Based on the rotation relationship, the transformation matrix of the inertial frame to the line-of-sight frame can be expressed as

$$L(q_\varepsilon, q_\beta) = \begin{bmatrix} \cos q_\varepsilon \cos q_\beta & \sin q_\varepsilon & -\cos q_\varepsilon \sin q_\beta \\ -\sin q_\varepsilon \cos q_\beta & \cos q_\varepsilon & \sin q_\varepsilon \sin q_\beta \\ \sin q_\beta & 0 & \cos q_\beta \end{bmatrix} \tag{A1}$$

According to the transformation between the inertial frame and line-of-sight frame, the projection of the predicted interception area in the target–interceptor impact plane can be expressed as

$$\overline{\mathcal{R}} = \left\{ (y,z) \middle| (y,z) \sim \mathcal{N}\left( \boldsymbol{y}, \boldsymbol{Q}_y \right) \right\} \tag{A2}$$

where

$$\begin{aligned} \boldsymbol{y} &= \begin{bmatrix} 0 & 1 & 0 \\ 0 & 0 & 1 \end{bmatrix} L(q_\varepsilon, q_\beta) \left( \begin{bmatrix} x_T(t_e) \\ y_T(t_e) \\ z_T(t_e) \end{bmatrix} - \begin{bmatrix} x_{s0} \\ y_{s0} \\ z_{s0} \end{bmatrix} \right) \\ \boldsymbol{Q}_y &= \left( \begin{bmatrix} 0 & 1 & 0 \\ 0 & 0 & 1 \end{bmatrix} L(q_\varepsilon, q_\beta) \right) \boldsymbol{Q}_{x_T} \left( \begin{bmatrix} 0 & 1 & 0 \\ 0 & 0 & 1 \end{bmatrix} L(q_\varepsilon, q_\beta) \right)^{\mathrm{T}} \end{aligned} \tag{A3}$$

where $\begin{bmatrix} x_{s0} & y_{s0} & z_{s0} \end{bmatrix}^{\mathrm{T}}$ is the coordinates of the initial line-of-sight frame origin in the inertial frame, $\begin{bmatrix} x_T(t_e) & y_T(t_e) & z_T(t_e) \end{bmatrix}^{\mathrm{T}}$ and $\boldsymbol{Q}_{x_T}$ are the mean value and covariance matrix of the predicted interception area in the inertial frame.

**Appendix B. Convergence Proof of the Area Coverage Optimization Algorithm**

Before the convergence proof of Algorithm 1, the definitions and lemma will be given first. Consider a general optimization problem as follows:

$$\begin{aligned} \min\ & g(x) \\ \text{subject to } & x \in \Omega \end{aligned} \tag{A4}$$

where $g(x)$ is a real-valued continuous function and $\Omega$ is the constrained set of $x$. Let $F$ be the set of optimal solutions of $g(x)$, and $\Gamma$ be an algorithm defined on $\Omega$. Denote $C$ to be a set, if $\forall x_0 \in C$ satisfies $\Gamma(x_0) \in C$, then $C$ is called the positive invariant set of $\Gamma$. A point $x_*$ is called a fixed point of $\Gamma$ if $x_*$ satisfies $\Gamma(x_*) = x_*$, and the set of $x_*$ is denoted by $B$. If the function $g(x)$ satisfies $g(\Gamma(x)) \leq g(x)$, $x \in C$ and the equality holds if and only if $x \in B$, then $g(x)$ is called a descent function of $\Gamma$.

**Lemma A1** [24] *If $g(x)$ is a descent function of the algorithm $\Gamma$, then the necessary condition that $x'$ is the minimum point of $x'$ is the minimum point of $g(x)$ is $x' \in B$, i.e., $\Gamma(x') = x'$.*

Now, we will give the convergence proof of Algorithm 1.

**Theorem A1** *Let $s(i) = \left[ \boldsymbol{p}_1^{\mathrm{T}}(i), \cdots, \boldsymbol{p}_N^{\mathrm{T}}(i) \right]^{\mathrm{T}}$, an iterative calculation of Algorithm 1, be denoted by $\Gamma$, then the relationship between $s(i+1)$ and $s(i)$ can be described by*

$$s(i+1) = \Gamma(s(i)) \tag{A5}$$

*If $B$ is a finite set of points, under the iterative calculation of $\Gamma$, $s(i)$ will converge to a point in $B$ when $i \to \infty$.*

**Proof of Theorem A1** For the $i$-th division of the predicted interception area, under the calculation of $\Gamma$, $s(i+1)$ satisfies

$$J(s(i+1)) \leq J(s(i)) \tag{A6}$$

Then, the performance index function $J$ is a descent function of $\boldsymbol{\Gamma}$.

Since the terminal point of interceptors are in the predicted interception area, $\boldsymbol{s}(i)$ is bounded, then there exists a subsequence of $\{\boldsymbol{s}(i)\}_{i=0}^{\infty}$ that converges to a point. Let $\{\boldsymbol{s}(i_k)\}$ be a converging subsequence, which converges to $\bar{\boldsymbol{s}}$. According to the continuity of $J$, $J(\boldsymbol{s}(i_k))$ converges to $J(\bar{\boldsymbol{s}})$. Next, it will be proved that for all subsequences, $\boldsymbol{s}(i)$ converges to $\bar{\boldsymbol{s}}$. According to the monotonicity of $J$ on $\boldsymbol{s}(i)$, it can be obtained that $J(\boldsymbol{s}(i)) - J(\bar{\boldsymbol{s}}) \geq 0$ $\forall i$. Based on the convergence property of $J(\boldsymbol{s}(i_k))$, for $\forall \varepsilon > 0$, there exists $i'_k$ such that $J(\boldsymbol{s}(i_k)) - J(\bar{\boldsymbol{s}}) < \varepsilon$ $\forall i_k > i'_k$. Then for $\forall i > i'_k$, $J(\boldsymbol{s}(i)) - J(\boldsymbol{s}(i'_k)) \leq 0$, thus we can obtain

$$J(\boldsymbol{s}(i)) - J(\bar{\boldsymbol{s}}) = J(\boldsymbol{s}(i)) - J(\boldsymbol{s}(i'_k)) + J(\boldsymbol{s}(i'_k)) - J(\bar{\boldsymbol{s}}) \leq \varepsilon \tag{A7}$$

i.e., $J(\boldsymbol{s}(i))$ converges to $J(\bar{\boldsymbol{s}})$ and $\{\boldsymbol{s}(i)\}_{i=0}^{\infty}$ converges to $\bar{\boldsymbol{s}}$.

Next, it will be proved that $\bar{\boldsymbol{s}} \in \boldsymbol{B}$. Assuming that $\bar{\boldsymbol{s}} \notin \boldsymbol{B}$, i.e., $\boldsymbol{\Gamma}(\bar{\boldsymbol{s}}) \neq \bar{\boldsymbol{s}}$, then there exists a constant $\varepsilon$ such that $\|\boldsymbol{\Gamma}(\bar{\boldsymbol{s}}) - \bar{\boldsymbol{s}}\| > \varepsilon$. According to the convergence property of $\boldsymbol{\Gamma}$, there exists $\delta > 0$ such that $\|\boldsymbol{\Gamma}(\bar{\boldsymbol{s}}) - \boldsymbol{\Gamma}(\boldsymbol{s}(i))\| \leq \varepsilon/3$ when $\|\bar{\boldsymbol{s}} - \boldsymbol{s}(i)\| \leq \delta$. Since $\{\boldsymbol{s}(i)\}$ converges to $\bar{\boldsymbol{s}}$, there exists $i_0$ such that $\|\bar{\boldsymbol{s}} - \boldsymbol{s}(i)\| \leq \delta$ when $i > i_0$, then we can have $\|\boldsymbol{\Gamma}(\bar{\boldsymbol{s}}) - \boldsymbol{\Gamma}(\boldsymbol{s}(i))\| \leq \varepsilon/3$. In addition, there exists $i_1$ such that

$$\|\boldsymbol{s}(i) - \bar{\boldsymbol{s}}\| = \|\boldsymbol{\Gamma}(\boldsymbol{s}(i-1)) - \bar{\boldsymbol{s}}\| \leq \varepsilon/3 \tag{A8}$$

if $i > i_1$.

Let $i_2 = \max\{i_0 + 1, i_1\}$, for $\forall i > i_2$, we can have

$$\|\boldsymbol{\Gamma}(\bar{\boldsymbol{s}}) - \bar{\boldsymbol{s}}\| \leq \|\boldsymbol{\Gamma}(\bar{\boldsymbol{s}}) - \boldsymbol{\Gamma}(\boldsymbol{s}(i-1))\| + \|\boldsymbol{\Gamma}(\boldsymbol{s}(i-1)) - \bar{\boldsymbol{s}}\| \\ \frac{\varepsilon}{3} + \frac{\varepsilon}{3} = \frac{2\varepsilon}{3} \tag{A9}$$

which contradicts with $\|\boldsymbol{\Gamma}(\bar{\boldsymbol{s}}) - \bar{\boldsymbol{s}}\| > \varepsilon$. Thus, $\boldsymbol{\Gamma}(\bar{\boldsymbol{s}}) = \bar{\boldsymbol{s}}$, i.e., $\bar{\boldsymbol{s}} \in \boldsymbol{B}$.

Since $\bar{\boldsymbol{s}}$ satisfies the minimum condition of $J$, then it can be seen that under the iterative calculation of Algorithm 1, $\{\boldsymbol{s}(i)\}_{i=0}^{\infty}$ will converge to the minimum point of $J$. $\square$

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
