# Peer review of "A Design Approach for Simultaneous Cooperative Interception Based on Area Coverage Optimization"

_drones, doi:10.3390/drones6070156_

Round 1

Reviewer 1 Report

1.     The paper is written well.

2.     In section 2.1, what is a problem to consider in dynamic conditions? does it able to consider the acceleration.

3.     For the Area coverage optimization algorithm, it’s only considered a spatial distribution, why is not consider the time change for static and dynamic conditions.

4.     Please elaborate on the introduction section.

5.     Please provide an explanation of the Figures in detail.

6.     Please check the typo error in the manuscript.

Reviewer 2 Report

The authors designed an area coverage optimization algorithm to realize cooperative interception. This paper discussed the relevant theoretical basis and carried out simulation experiments, which had certain application value. I recommend the acceptance of this paper after revision. The following modifications are suggested.

1. Lines 93 to 100 should be deleted.

2. The writing of this paper needs to be greatly improved, and there are many problems in form and grammar.

3. The descriptions in section 2 and 3 are not standardized, especially the explanation of the formula is not clear, which affects reading and understanding.

4. The typesetting of this paper is rather sloppy, especially the format and symbols of the formula are not standardized. A lot of symbols are omitted, (e.g. Line 122,126,150).

5. The quality of the diagram needs to be improved, such as Figure 2.

6. In this paper, the real-time problem is mentioned, but there is no further analysis of the real-time performance of the designed algorithm, and there is a lack of comparison with other algorithms. This is option for authors to conder whether it need to be revised.

7. In the process of solving optimization problems, why not use classical mathematical optimization methods or intelligent optimization methods (such as evolutionary computation)? It is suggested to analyze and explain in the text.

8. In oder to broaden the scope of this paper, the authors may introduce more optimization algorithms related to coverage optimization such as

  8.1  “Node Coverage Optimization Strategy Based on Ions Motion Optimization”, Journal of Network Intelligence, Vol. 4, No. 1, pp. 1-9, Feb 2019

  8.2 An Optimal Node Coverage in Wireless Sensor Network Based on Whale Optimization Algorithm,” Data Science and Pattern Recognition, vol. 2(2), pp. 11-21, 2018

    8.3 A Coverage Loopholes Recovery Algorithm in Wireless Sensor Networks, Journal of Information Hiding and Multimedia Signal Processing, Vol. 7, No. 6, pp. 1354-1364, November 2016

9. The authors may consider to supplement comparative experiments with other representative algorithms.

Reviewer 3 Report

This paper deals with the problem of simultaneously cooperative interception using multiple interceptors.

It is a very extensive article, complex, with a thorough mathematical study and, in my opinion, the subject it addresses is really interesting, especially in the current circumstances where military applications of UAVs are presenting an important development. I think it presents enough contributions to be accepted, but I find some aspects confusing.

In my professional experience prior to my incorporation to the University I worked for some years in the European defense sector for NATO (present Thales Group) and one of our works was oriented to the design of anti-missile systems for vessels that involved localization and detection of the threat, prediction of its trajectory and interception. There is one assumption the authors make that I don't understand and one piece of data I can't seem to find.

Assumption 1: The relative trajectories between an interceptor and the target can be 109 linearized with respect to the initial line of sight, and the acceleration of interceptor is 110 perpendicular to the line of sight

It seems that possible trajectory changes are not considered, I think this point should be clarified.

It is not stated how the position of the target would be determined. That is, no indication is given to the reader as to what systems would be employed to detect the intrusion.

I found the part that the authors define as experimental verification very difficult to understand and very confusing. I understand that all the results are simulations. I think this section should be rewritten clarifying exactly what is being presented and, if possible, a real case would be ideal to demonstrate the effectiveness of the algorithm.

The run time of the algorithm is also not stated, and I believe it is relevant.

Table 1 indicates velocities in the order of 300 m/s for the target which are close to 1 Mach. Maybe it is a mistake on my part, but I do not understand a so high speed. Military drones like the Reaper have VNEs that do not exceed 500 km/h. I think this point should be clarified from the beginning of the article by stating which target and which interceptor would be used with this algorithm.

Finally, it would be desirable to improve some graphics whose fonts are somewhat small or blurred in the conversion.

In any case, I congratulate the authors for the complexity of the work done.

Round 2

Reviewer 1 Report

The authors have addressed all my concerns, and this paper looks good now.